# Experiences of Women Who Refuse Recall for Further Investigation of Abnormal Screening Mammography: A Qualitative Study

**DOI:** 10.3390/ijerph19031041

**Published:** 2022-01-18

**Authors:** Wei-Ying Sung, Hui-Chuan Yang, I-Chen Liao, Yu-Ting Su, Fu-Husan Chen, Shu-Ling Chen

**Affiliations:** 1Department of Family Medicine, Taichung Veterans General Hospital, Taichung 40705, Taiwan; swy2015@vghtc.gov.tw; 2Department of Nursing, Hungkuang University, Taichung 43304, Taiwan; jiuan@hk.edu.tw (H.-C.Y.); micky26@sunrise.hk.edu.tw (I.-C.L.); ytsu67@hk.edu.tw (Y.-T.S.)

**Keywords:** breast cancer, mammography, screening, follow-up, qualitative study

## Abstract

Breast cancer has the highest incidence among all cancers for women in Taiwan. The current screening policy in Taiwan provides biennial mammogram tests for all women aged 45 to 69 years. A recommendation for further investigation is sent via post to women with a BI-RADS result of 0. The proportion of women who followed-up with a recall request for further investigation after an abnormal mammogram has been below 92.5% in recent years. Therefore, we aimed to explore the experiences of these women who refused recall for further investigation despite an abnormal mammogram. Purposive sampling was conducted on 13 women who refused recall for further examination of abnormal screening mammograms. Data collection included inductive, in-depth interviews or telephone interviews. A content analysis was applied. Three themes were identified: (1) negative screening experiences, (2) struggling with ’to go or not to go’, and (3) rationalizing without a follow-up examination. The first theme included three subthemes: (1) pain of examination, (2) the inconvenience of medical treatment; waiting, and (3) dissatisfaction with having to pay for further examination. The second major theme included three subthemes: (1) perceiving one’s susceptibility to breast cancer as very low, (2) questioning the accuracy of the results, and (3) procrastinating with a “head-in-the-sand” mentality. The third major theme included two subthemes: (1) fatalism and (2) paying attention to self-cultivation. In conclusion, the findings provide important information to healthcare providers involved in case management related to the actual living experiences of women with abnormal screening mammogram results and the additional education required to raise breast cancer awareness in the general public to achieve overall caring goals.

## 1. Introduction

Breast cancer is the most common cancer in women worldwide. In 2020, there were 2.3 million women diagnosed with breast cancer and 685,000 deaths globally [1]. In Taiwan, where breast cancer has the highest incidence among all cancers in women, the incidence of breast cancer in 2018 was 78.86/100,000 [2,3], and Taiwan’s rate was also the highest among Asian countries [4]. The mortality rate of breast cancer in Taiwan in 2018 was 12.34/100,000 [3]. The screening policy in Taiwan, since 2002, has recommended a biennial mammogram for women aged 45 to 69 years and for those aged 40 to 44 years with a family history of breast cancer [3]. A recommendation for additional testing is sent via post to women with BI-RADS results of 0, 3, 4, or 5 [3]. Especially for BI-RADS 0, it indicates incomplete—additional imaging evaluation and/or comparison to prior mammograms is needed [3,4]. At present, mammography has been recognized as the best breast cancer screening tool in the world, and it is the most common tool used worldwide for breast cancer screening [4,5,6,7,8]. 

Complete and effective screening must be carried out when abnormal results are found in a public screening. That is, patients with abnormal screening results must continue to receive follow-up diagnosis and follow-up treatment arrangements to achieve the purposes of early detection and early treatment of cancer. According to the Health Promotion Administration of the Ministry of Health and Welfare in Taiwan, about 1,470,786 women received a screening mammography in 2018–2019, which is a screening rate of 40% [9]. However, about 92.3% of the cases with positive breast cancer screenings (BI-RADS 0, 3, 4, or 5) received a follow-up examination, leaving 7.7% of women with a high risk of breast cancer lost to follow-up [9]. Therefore, for early detection of breast cancer, it is of paramount importance to increase the follow-up rate in women with abnormal screening mammogram results [2,3].

Several studies have proposed methodologies to promote screening mammography [10,11,12], to use multiple approaches such as behavioral and cognitive combined approaches [13], and to understand women’s benefits and barriers to screening mammography [14,15]. Based on these above studies, behavioral interventions, theory-based education, and interventions targeted at physicians or providers have been effective in promoting screening. However, few studies have focused on the causes of loss to follow-up in women with abnormal screening mammography results. When a woman receives a report of an abnormal result after a mammography examination and learns that further examination is needed to confirm the diagnosis, she may wait to respond due to the psychological distress caused by the threat of cancer [16]. The physical and psychological health, and even the quality of life, of the patient can be affected for weeks to months or even years, and responses vary from person to person [17]. When an individual cannot accurately predict what will happen, the feeling of uncertainty can upset the order of daily life, threaten the individual’s feelings, or challenge the belief that the individual is healthy. However, it is uncomfortable to be in a state of uncertainty [18], and this discomfort manifests as different levels of psychological distress, which may include emotional instability, worry, fear, depression, fear, shock, weakness, anxiety and restlessness [6,19,20,21]. Therefore, this study used qualitative research methods to understand the status of women who have not sought follow-up treatments after breast cancer screening. Through this research, it is hoped that medical staff will be able to increase the prevalence of breast cancer screening. This research is expected to provide education regarding breast cancer awareness to the general public, and medical personnel will play a vital role in improving abnormal screening mammography follow-up rates.

## 2. Methods

### 2.1. Design

A qualitative study approach was used to understand the experiences and individual perspectives of women, as well as their attitudes, situations, expectations, and suggestions for refusing the recall for further investigation of abnormal breast cancer screening results. Purposive sampling was used to recruit women who met the following criteria: (1) screening mammography at the Health Promotion Administration, Ministry of Health and Welfare, in 2016–2017 with a BI-RADs category result of 0, (2) refusal of recall for further examination of abnormal breast cancer screening, and (3) voluntary agreement to participate in this study. All study participants provided informed consent during individual or telephone interviews. The exclusion criterion was a BI-RADs result of 4.

### 2.2. Data Collection

Data collection consisted of semi-structured interviews to determine the participant’s reasons for refusing the recall for further investigation of an abnormal breast cancer screening result based on the following questions: (1) What did you think when you were informed about an abnormal breast cancer screening report via postal mail or a telephone call from the nurse? (2) So far, since receiving the abnormal report, you have not had a follow-up examination. How do you deal with it? (3) How did the nurse communicate with you and explain the abnormal report? Do you have any suggestions that would make more women willing to return to further examinations? and (4) Based on your mammogram screening experience, how do you think the service can be improved? Data collection was conducted from October to December 2018. None of the participants withdrew from the study. Data collection was completed using the data saturation criterion through the repetition of information in the statements from a total of 12 telephone interviews and one in-depth interview. The interviews ranged in length from 45 to 60 min.

### 2.3. Analysis and Rigor

All audio-recorded interviews were transcribed verbatim and analyzed by all authors using content analysis [19]. The analysis followed five steps: (1) repeatedly listening to audio-recorded interviews and repeatedly reviewing all transcripts to obtain an overall understanding of and familiarity with the data; (2) highlighting and marking important experiences in the transcripts; (3) searching for meaning units in the transcripts; (4) developing an initial list of ideas about the data, from which initial codes are generated; and (5) reviewing the initial codes to discover the themes [22]. To ensure the trustworthiness of the study, our procedure followed the guidelines proposed by Lincoln and Guba [23]. Credibility of the data was enhanced by the authors’ expertise in nursing and qualitative research, allowing us to fully understand the women’s experiences regarding the refusal of recall for further investigation of abnormal breast cancer screening. Transferability was facilitated through the use of purposive sampling, and the data sources were enriched by including participants of different ages and levels of education. Dependability was promoted by the authors meeting frequently to discuss the data analysis and by checking and rechecking the labeling, sorting and naming of themes during data analysis for verification as suggested. Confirmability was ensured by describing the entire research process and procedures in detail, keeping a reflexive journal, and maintaining an audit trail [23]. Women’s experiences were extracted as thick descriptions of the related phenomena.

### 2.4. Ethical Considerations

This study was performed in accordance with the principles of the Declaration of Helsinki. All the study procedures were approved by the Hospital Human Investigation Committee at each of the medical centers (IRB No. CE18055B). The participants’ rights of anonymity, confidentiality and withdrawal from the study were explained at the time of the interview. 

## 3. Results

### 3.1. Participants’ Characteristics and Identified Themes

Table 1 shows participants’ sociodemographic characteristics. The mean age of the participants was 59.5 years (range: 49–68). The women had educational backgrounds of less than senior high school (38.5%) and greater than senior high school (61.5%). All women were married (100%) and had one to two children (53.8%) or more (30.7%).

During the analysis of the interviews, three specific themes emerged: (1) negative screening experiences; (2) struggling with ’to go or not to go’; and (3) rationalizing without follow-up examination (Table 2).

### 3.2. Theme 1. Negative Screening Experiences

This theme refers to a series of negative reactions to the prior examination when a woman receives a positive notification of breast cancer screening. When women express their own screening experiences, they will recall the overall process of screening and follow-up. 

#### 3.2.1. Pain of Examination

A mammogram is an X-ray examination of the breast. About 53.8% of women may feel pain during the procedure, while others may feel the pain is tolerable. Each breast fits onto a flat X-ray plate. A compressor then pushes the breast down to flatten the tissue. This provides a clearer picture of the breast.

As the 64-year-old case G exclaimed: “It was so painful! I don’t know why this examination has to squeeze my breast so bad! This is terrible! I don’t like it!”

As 58-year-old case M stated, 

“I was very uncomfortable during the examination. It was terribly painful, and it hurt constantly for two weeks. It really hurt! It was really painful. I really don’t want to do this examination ever again!”

If provided options, they would refuse to do a mammography, as 68-year-old case A said: “If you want me to do that squeezing examination (mammography), you don’t need to call me anymore! It was very painful!”, “This brings me back to the very first time, when I experienced the same pain!”

#### 3.2.2. Inconvenience of Medical Treatment: Waiting Time

When women receive a positive breast cancer screening notification, they often recall their negative experiences of the past, especially the time spent waiting for the examination at the hospital. For example, 66-year-old case B, who had had colon cancer more than ten years previously, mentioned:

“I don’t want to go to the hospital anymore, I’m so scared!…I waited and waited the whole morning for the doctor to see me for only 2 or 3 min…”

Some patients choose to give up because of the long waiting times at the hospital. For example, 58-year-old case M said: 

“I have registered at the other General Hospital for the further check-up (ultrasound examination), but it requires a lot of steps and time to wait until my turn in the afternoon, so I did not want to wait anymore.”

As 68-year-old case A, who is illiterate and lives alone, said:

“The mobile mammography vehicle is very convenient for time! Normally, you have to go to the hospital for a long time to queue up for this examination. Visiting the hospital for an examination is a lot of trouble for me.”

#### 3.2.3. Dissatisfaction about Paying for Further Examination

Unlike the screening, which was free, women who were recalled for further examination of an abnormal breast cancer screening have to pay their own medical bill. Some participants expressed strong dissatisfaction with this situation. For example, case G said:

“Breast cancer screening is a free service. Why should I have to pay for further examination with my own money? This is terrible. Plus, during the mammography, the squeezing hurt so much; it was very uncomfortable. And now I need to pay for an examination!?! I can’t accept it. Who would be willing to do it?”

If the further examination cost them money and their personal time, they would choose not to attend the follow up. As noted by 56-year-old case L:

“You keep calling and telling us to go back for further examination. It makes people wonder if there is any problem. So I try to cooperate with you and follow your instructions, but the problem is…I have to pay for this on my own…do you know what I mean? I’m not happy about this.”

### 3.3. Theme 2. Struggling with ‘To Go or Not to Go’

After receiving an abnormal breast cancer screening report, a woman will subjectively consider her own or others’ medical experiences, such as consciously thinking it should be fine, being asymptomatic, having no family history of breast cancer, and finding no lumps during self-examination. They may even consider the medical report a false positive. All these reasons support “not going” for a further examination. Medical experiences and treatments that might support “going” for a further examination are awareness of breast cancer threats and others’ experiences.

Struggling with a decision on whether ‘to go or not to go’ to a hospital for further examination can lead to repeated and tangled thoughts. When faced with the question of ‘to go or not to go’ for further examination, there are three kinds of psychological struggles, which are as follows:

#### 3.3.1. Perceiving One’s Susceptibility to Breast Cancer as Very Low

After receiving notification of an abnormal report, some women may not return to the hospital in time for a further examination for several reasons. These reasons may be that they do not feel any threat to their health, their own self-examinations revealed no lumps or hard masses, they have no abnormal symptoms, they feel healthy, and they have no family history of breast cancer. As a result, they do not return to the hospital in time for a follow-up.

A total of ten of the thirteen (76.9%) participants reported that they did not have any abnormal breast cancer symptoms currently. Two had a history of benign breast-related disease and one had had colorectal cancer.

Case D viewed the abnormal report as not serious. She said, “I didn’t feel anything hard…nothing. I think I’m fine. Also, I’ve received a recall notification before, so I’m not nervous at all.”

Case E said, “Because I have done it before, there are reports of abnormalities just like before, so I am not nervous at all. It is no different from before.”

Although Case B had a ten-year history of colorectal cancer, she was not willing to be re-examined this time. She stated that the examination more than a year before had been benign and she did not want to be re-examined.

Case C felt that she usually conducted self-examinations without feeling any lumps, and said, “I did not return to the hospital for re-examination because I don’t think there is anything, and I can’t feel anything! When I take a bath, I check them. Breasts sometimes have size, not necessarily that there is a problem with size!”

About three quarters (76.9%) of the women expressed that they had neither symptoms nor discomfort, so they did not want to be further examined. As case M said, “However, I have no symptoms yet, and so far there is no discomfort…” Although a person with a family history of breast cancer should be regarded as having a high risk of breast cancer, two women whose sisters had had breast cancer still consciously felt that they were in good health and did not have cancer.

#### 3.3.2. Questioning the Accuracy of the Results

Many participants wondered if, with so many cases receiving a follow-up consultation report, was there a real problem? For example, Case D thought, “Is there really a problem or something? Then why didn’t you tell me directly? If there was really a problem (here and now during the check), why can’t we just talk about it directly? I also question whether there is something wrong with the screening process or the machine itself, and it’s not that our bodies are abnormal”. 

Case L said, “It may be that the scanning process of your mobile mammography vehicle is not as good! So it may have these phenomena. And the scanning time is very long, and the result is still not good. I don’t want to do mammography the next time.”

Case G also had the same experience. She claimed that “the machine did not do it accurately.”

#### 3.3.3. Procrastinating Ostrich Mentality

After receiving the notification of abnormal breast cancer screening, women may consult their family members for their opinions and ideas, and they will also refer to the similar experiences of others around them to gauge whether they want to be followed up and return to the hospital for re-examination. Women eventually chose not to undergo a follow-up examination.

As Case D stated, “I asked my classmate whose wife works in a hospital. He said you still have to check it, but I think it is probably a lipoma, so it’s not that serious.”

Case K, whose husband had been urging her to go back to the clinic for re-examination, said, “My husband has been pushing me. He urged me three times and four times”. Then she said, “I also know in my heart that it must be checked. I would like to say something bad, and I would also like to say that it probably is not that serious, anyway, it is a contradiction!”

### 3.4. Theme 3. Rationalizing without Follow-Up Examination

When the women did not follow up and get re-examined, they would adopt either fatalism or a health regimen, and their own prescriptions, to allow themselves to accept and face it rationally.

#### 3.4.1. Fatalism

Whenever people face an unexplainable life phenomenon, they naturally choose a rational explanation, and they may be accustomed to attributing events to fate. Such a mindset can allow them a certain degree of comfort when encountering possible abnormal breast screening results. To alleviate the suffering in life and maintain a calm happiness, women will adopt “fatalism” to explain their own outlook on life. Their fatalism includes the ideas that *life is impermanent* and everything is *bearish*; life and death are set by fate, and one should let nature take its course.

As Case C pointed out, “Everyone will die. Don’t be afraid. When you are happy, our condition will not worsen. It will repair our immune system. And everyone has cancer cells in their bodies, it’s just whether they have attacked or not.”

The women mentioned that life and death are natural phenomena. People cannot control life and death, or whether they become sick or not. It is an arrangement of God’s will. Therefore, excessive fear, worry, and even anxiety are useless. It will increase your troubles and unhappiness. Case A, who had the same experience, also said, “You have to leave the world when the time is up anyway.” Case J said, “If I have (physical symptoms), then the health check will be fine. If I really have cancer, it will be life and death.” In addition, Case A still hasn’t been re-examined and said, “I see it very openly. People come to live in the world. It’s useless to think so much. I think it’s good to pay attention to body care and exercise.”

#### 3.4.2. Paying Attention to Self-Cultivation 

The participants used folk remedies suggested by their family members and started taking them after the breast screening to avoid breast cancer in the upper body, or they used their own health care methods, diet choices, exercise and other methods in order to maintain their health.

Case A said, “I myself take Chinese medicine to maintain my body. When I received a positive follow-up call after mammography, I bought Chinese medicine to take, and I felt it was effective after taking it.” Case C mentioned the anti-cancer lifestyle, or living happily. She said, “Why haven’t so many cancers attacked? This means that he may live his life happily. So a person’s mood is very important.” Case M used Chinese medicine to maintain her health, saying, “Because my family is also a family of Chinese medicine, I know the prescriptions for cancer. There are breast cancer prescriptions for breast cancer and also prescriptions for colorectal cancer. I have relied on Chinese medicine to regulate my body since I was a child.”

## 4. Discussion

To our knowledge, this is the first study to adopt a qualitative approach to explore the experiences of women who refused recall for further investigation of abnormal screening mammography results. Many studies have discussed the psychological consequences of false-positive screening mammograms [24,25,26,27]. However, none have specifically focused on the experiences of such women. There are similarities and differences between previous studies and ours. Our study showed that women cite their own negative screening experiences, including dissatisfaction with the screening process, pain during the examination, the waiting time and the experience of the process for further visits, and previous examination results showing pseudo-abnormalities. Such negative experiences can make women fear re-examination, or even hesitate or refuse it. This result is similar to the results of a systematic review by Brett et al. [28], and it also echoes the findings of the qualitative research of He et al. [29], which recommended payment of insurance benefits, convenient inspection arrangements, and less painful screening procedures. Such factors can affect whether a woman will be subject to regular inspections or further re-examinations. This is also consistent with the argument put forward by Kuo et al. [2]; that is, unlike foreign private insurance, Taiwan provides universal health insurance coverage, which will affect people’s screening and subsequent medical treatment. 

According to the experience of the screening site, since mammography is an imaging examination that uses low-dose x-rays to image the breast tissue, the mammogram is taken through a special machine that involves compression and squeezing. This examination process causes pain and discomfort, but it can be used to detect calcification points or tiny tumors in the breast, or to find asymptomatic zero-stage breast cancer. Something similar was reflected in the study by Kornguth et al., who found that a large proportion (up to 91%) of women report experiencing some degree of pain during mammography [30]. Although the pain is generally in the low to moderate range, a portion of women did report moderate to intense pain [21,30,31]. Another study found that 77% of women reported that the procedure was painful. Of those reporting pain, 60% described the pain intensity as moderate or severe. In 67%, the pain resolved within 10 min [32]. Likewise, in our study, most of the women did not understand why the breasts were squeezed. They did not understand the reason for using this painful inspection method instead of a painless inspection method, such as ultrasonic inspection. Our study also collected suggestions on how to strengthen the effective promotion of breast cancer screening concepts so that women have an accurate understanding and concepts about breast cancer screening, which is one of the important topics for breast cancer screening in the future.

When conducting a telephone follow-up, the nurse must first explain the costs that may be incurred during the re-examination and return to the clinic, and each must be clearly explained. If the patient does not return to the hospital for re-examination and follow up, the nurse will actively provide “referral” related tasks to assist the patient to go to the medical institution of her choice or near her home for reexamination and tracking. After that, the patient can take the current mammography CD-ROM sent by the hospital with the referral triplicate to the medical institution of choice for reexamination. In this way, the patient can also reduce her medical expenses (a partial burden reduction or registration fee reduction). In addition, it may also reduce the cost of treatment. It takes a long time to go to the medical center for treatment. In addition, it is important to consider whether there are other, friendlier medical treatment measures that can replace the exemption measures, reduce the negative feelings of the patient, and increase the willingness of the patient to return to the hospital for review and follow up.

The pull between “not going” and “going” for a re-examination shows that the perceived threat of breast cancer is low, which is similar to the results of Kuo et al. [2], if the mammography result is BI-RADS 0. Likewise, in our study, the most likely reason for not receiving follow-up treatment was “thinking she was in good health and did not need to accept follow-up treatment”. This thinking persisted even though two women had had benign breast-related diseases; one had had colorectal cancer and two had family histories of breast cancer. Another study explored the factors associated with annual-interval mammography for women aged 40 to 49 years. The results showed that factors related to nonadherence included the lack of knowledge/not thinking mammograms are needed, the cost, being too busy, and forgetting to make/keep appointments [21,33]. Likewise, in our study, the factors related to not receiving follow-up included negative screening experiences, which included three categories: the pain of examination; the inconvenience of medical treatment, including the waiting time; and dissatisfaction with having to pay for a further examination. During the interview, it was found that even if the patients felt nervous upon receiving the notification of the abnormality, the absence of any particularly uncomfortable symptoms would lead them to believe that there was no need for re-examination. Thus, they did not go for the examination immediately. In contrast, He et al. [29] found that women agreed that regular mammography was necessary for the early detection of breast cancer and were willing to accept follow-up suggestions from professional medical personnel.

Regarding rationalizing without follow-up examination, when women do not attend a follow-up examination, they may adopt fatalism and pay attention to self-cultivation methods so that they can accept and face their disease rationally. Chinese fatalism is the most common, including the impermanence of life. This is the same as the findings in other research in Taiwan [21,34]. This also explains why, when women do not attend a follow-up, they will take care of themselves. For example, in the case of this study, they used diet and food, religious or ritual support, Chinese medicines and herbs, and exercise to adjust their breathing. These methods of rationalizing acceptance and confrontation used by the participants in the current study are exactly the folk remedies pointed out in the literature.

## 5. Study Limitations

This study had several limitations. The first was the small sample size. Only 30 women met the inclusion criteria. Of these, only 13 women were willing to be interviewed by telephone, and they were not willing to give face-to-face interviews. Only one participant was willing to meet face-to-face for the interview. The second limitation was the interview mode, which may have affected data collection. It is not easy to detect non-verbal expressions and body language in a telephone interview, and it is also easy to be interrupted by factors in the surrounding environment. The final limitation was that we excluded BI-RADS 4 follow-up cases, because different treatments produce different levels of anxiety and experiences. Future studies could incorporate BI-RADS 4, and even BI-RADS 3 and 5, to understand why the women did not undergo further follow-up examinations.

## 6. Conclusions

The study results showed that women with abnormal mammography screening but no follow-up examination perceived the risk of breast cancer as low (76.9% of respondents) and found the examination painful (53.8% of respondents). Therefore, we need to improve their understanding of breast screening knowledge and reduce the physical pain of the procedure.

The women also expect to be understood and supported, which is also a possible factor that promotes their willingness for follow-up. If screening nurses can demonstrate effective professional communication skills and understand their feelings, thoughts and attitudes, it will help nurses to improve their understanding of these women and how to reduce their negative experience as a strategy to increase follow up visits. 

## Figures and Tables

**Table 1 ijerph-19-01041-t001:** Characteristics of the participants (*N* = 13).

Code	Age	Education	Number of Children	Breast Related Diseases	Family History of Breast Cancer	Mobile Mammography Vehicle	Abnormal Mass
A	68	Illiterate	5	No	sister	Yes	No
B	66	Secondary	3	colorectal cancer	No	Yes	Yes
C	67	Secondary	0	No	No	Yes	No
D	49	Higher education	3	benign	No	No	No
E	55	Higher education	2	benign	No	No	Yes
F	57	College/University	2	No	No	No	No
G	64	College/University	2	No	No	No	Yes
H	64	College/University	0	No	No	No	No
I	54	College/University	1	No	sister	No	No
J	64	Elementary	6	No	No	Yes	No
K	51	Higher education	2	No	No	No	No
L	56	College/University	2	No	No	Yes	No
M	58	Higher education	2	No	No	Yes	No

**Table 2 ijerph-19-01041-t002:** Themes and Sub-themes.

Theme	Sub-Theme	Number (Percentage)
Negative screening experiences	Pain of examination	7(53.8%)
	Inconvenience of medical treatment: waiting time	4(30.8%)
	Dissatisfaction with having to pay for a further examination	2(15.4%)
Struggling with ’to go or not to go’	Perceived susceptibility to breast cancer very low	10(76.9%)
	Questioning the accuracy of the results	3(23.1%)
	Procrastinating ostrich mentality	4(30.8%)
Rationalizing without follow-up examination	Fatalism	5(38.5%)
	Paying attention to self-cultivation	4(30.8%)

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
