# Peer review of "Experiences of Women Who Refuse Recall for Further Investigation of Abnormal Screening Mammography: A Qualitative Study"

_ijerph, 2022, doi:10.3390/ijerph19031041_

Round 1

Reviewer 1 Report

Authors investigated the reasons why women refuse recall after abnormal screening mammograms through semi-structured interviews. In this way, they hope to improve the experience of these women and to raise awareness about breast cancer.

  • The major limitation of the study is the small cohort composed of 13 women with a BI-RADS 0 result and with mammograms performed in 2016-2017. In fact, it is not clear why authors excluded women with BI-RADS 4 that receive the same recommendation for further investigation, as described in the Introduction. Moreover, a longest time interval should be also considered to overcome the limitation of the small sample size.
  • The percentage of woman who refuse recall in Taiwan should be specified in the Introduction.
  • Authors should be more specific about the exclusion criteria in section 2.1.
  • Authors should improve the description of the results, i.e. section 3.2-3.3-3.4. They should reduce the number of reported sentences to improve readability, leaving the most significant ones and summarizing the main concepts reported in the others. Eventually, they can create a supplementary file with all the removed sentences.
  • Conclusions are too long and need to be summarized.
  • Please add recent references.
  • Table 2: please add percentage/number of women reporting each sub-theme to understand the most common reason for refusal.

Author Response

Author's Reply to Reviewer

Comment 1. The major limitation of the study is the small cohort composed of 13 women with a BI-RADS 0 result and with mammograms performed in 2016-2017. In fact, it is not clear why authors excluded women with BI-RADS 4 that receive the same recommendation for further investigation, as described in the Introduction. Moreover, a longest time interval should be also considered to overcome the limitation of the small sample size.
Answer: Thank you for your comments. In Taiwan, according to the hospital's preventive health care policy for mammograms, the mammogram results of BI-RADS 0 and BI-RADS 4 cases have different re-examination and diagnosis procedures. BI-RADS 0 indicates Incomplete – Additional imaging evaluation and/or comparison to prior mammograms is needed. BI-RADS 4 means Suspicious abnormality – Biopsy should be considered. Therefore, a BI-RADS 0 case needs to return to the hospital for ultrasound examination within 3 months, while a BI-RADS 4 case must return to the hospital for tissue biopsy within 3 months. Since different re-examination and diagnostic procedures are associated with different degrees of anxiety and experience, this study is based on the homogeneous BI-RADS 0 as the main research participants.

Comment 2. The percentage of woman who refuse recall in Taiwan should be specified in the Introduction.

Answer: We agree and appreciate your suggestion. (page 2, lines 44–49)

Comment 3. Authors should be more specific about the exclusion criteria in section 2.1.

Answer: Thank you for your comments.
As in mammography, BI-RADS 0 and BI-RADS 4 and 5 have different review and diagnostic procedures for mammogram results. For example, BI-RADS 0 means return within 3 months for ultrasound, while BI-RADS 4 and 5 patients must return within 3 months for tissue biopsy. Due to the varying levels of anxiety and experiences with different review and diagnostic procedures, this study focused on homogeneous BI-RADS 0.

Comment 4. Authors should improve the description of the results, i.e. section 3.2-3.3-3.4. They should reduce the number of reported sentences to improve readability, leaving the most significant ones and summarizing the main concepts reported in the others. Eventually, they can create a supplementary file with all the removed sentences.

Answer: We agree with you and decided to delete some reported sentences. In the description of the results, the reported sentences by cases have been deleted in the revised manuscript. (page 4, lines 139-page 7 line 293)

Comment 5. Conclusions are too long and need to be summarized.

Answer: Thank you for your comments. The conclusions have been condensed in the revised manuscript. (page 9, lines 385-395)

Comment 7. Table 2: please add percentage/number of women reporting each sub-theme to understand the most common reason for refusal.

Answer: We agree and appreciate your comment. Table 2 now includes a percentage/number column in the revised manuscript. (page 4, lines 138)

Reviewer 2 Report

  1. Abstract needs modification, please include the basic findings of the study.
  2. Please include latest information, data include latest stats of 2020 if possible.
  3. The data is from a very limited source, which affects the study source.
  4. I expect graphical presentation or pie chart of the study, so that readers may find it more appropriate.
  5. Multiple grammatical errors in the whole manuscript.

Author Response

Author's Reply to Reviewer

Comment 1. Abstract needs modification, please include the basic findings of the study.
Answer: Thank you for your comments. We have modified the abstract and added the basic findings in the revised manuscript. (page 1, lines 18-22)

Comment 2. Please include latest information, data include latest stats of 2020 if possible.
Answer: Thank you for your comments. We have added the latest information in the revised manuscript. (page 10, lines 414,430,432, and 458)

Comment 3. The data is from a very limited source, which affects the study source.

Answer: Thank you for your comments. We also write on research limitations. (page 9, lines 373-384)

Comment 4. I expect graphical presentation or pie chart of the study, so that readers may find it more appropriate
Answer: We agree and appreciate your comment. We have added table 2 now includes a percentage/number column in the revised manuscript. (page 4, lines 138)

Comment 5. Multiple grammatical errors in the whole manuscript.

Answer: Thank you for your comment. We have invited Americans to edit manuscript.

Reviewer 3 Report

This is a very nicely conducted study exploring the experiences of women who refused recall for further investigation, despite an abnormal mammogram. This qualitative research study was carried out with a purposive sampling and included inductive, in-depth interviews or telephone interviews. The topic is of interest and meaningful, the moethodology is robust, contenti s scientifically sound and the draft is clearly written. I woud favour publication. Thnaks for the opportunity to review this manuscript.

I would suggest the authors to:

  • With respect to the theme of ‘waiting time’ and ‘dissatisfaction about paying for further examination’, I would suggest the authors to elaborate within the discussion about the structure of Taiwan healthcare system in terms of organization, access and delivery of services and the influcence it may have on the answers reported by the women interviewed.
  • I would suggest the authors to address the following potential biases in the limitation section: sampling bias, non-response bias, order bias, recall bias, response bias and point out how the methodology used in the present study tried to overcome these potential limitations.

Author Response

Author's Reply to Reviewer

This is a very nicely conducted study exploring the experiences of women who refused recall for further investigation, despite an abnormal mammogram. This qualitative research study was carried out with a purposive sampling and included inductive, in-depth interviews or telephone interviews. The topic is of interest and meaningful, the moethodology is robust, content is scientifically sound and the draft is clearly written. I woud favour publication. Thnaks for the opportunity to review this manuscript.
Answer: Thank you so much for your encouraging remarks.

Comment 1. With respect to the theme of ‘waiting time’ and ‘dissatisfaction about paying for further examination’, I would suggest the authors to elaborate within the discussion about the structure of Taiwan healthcare system in terms of organization, access and delivery of services and the influcence it may have on the answers reported by the women interviewed.

Answer: Thank you for your comments. In Taiwan, mammography provides preventive care and is free, while follow-up re-examination and diagnosis must be paid to see the doctor for examination. General medical institutions charge registration fees and related inspection fees in accordance with health insurance regulations.

Comment 2. I would suggest the authors to address the following potential biases in the limitation section: sampling bias, non-response bias, order bias, recall bias, response bias and point out how the methodology used in the present study tried to overcome these potential limitations.

Answer: Thank you for your comments.
To ensure the trustworthiness of the study, our procedure followed the guidelines proposed by Lincoln and Guba (1985) : credibility, transferability, dependability and confirmability. (page 3, line 107-118). It should be possible to achieve the bias of the quantitative research proposed by the reviewer. If not, we also write on research limitations. (page 9, lines 373-384)

Round 2

Reviewer 1 Report

Authors improved the readability of the manuscript, especially in the Result section. They could modify few details more:

Line 38: A deeper explanation of additional diagnostic procedures according to the BI-RADS category should be added, especially for BI-RADS 0. (E. g. BI-RADS 0 indicates Incomplete – Additional imaging evaluation and/or comparison to prior mammograms is needed etc.)

Lines 82, 85: BI-RADS instead of BI-RAD

Table 2: you should change the layout of the last column – e.g. number (percentage)

Author Response

Comment 1: Line 38: A deeper explanation of additional diagnostic procedures according to the BI-RADS category should be added, especially for BI-RADS 0. (E. g. BI-RADS 0 indicates Incomplete – Additional imaging evaluation and/or comparison to prior mammograms is needed etc.)
Answer 1: Thank you for your comments. We have added explanation in the revised manuscript. (page 1, lines 38-40)

Comment 2 : Lines 82, 85: BI-RADS instead of BI-RAD
Answer 2: Thank you for your comments. We have revised in the revised manuscript. (page 2, lines 84, 87)

Comment 3 : Table 2: you should change the layout of the last column – e.g. number (percentage)

Answer 3: Thank you for your comments. Table 2 have been changed in the revised manuscript. (page 4, lines 138-139)

Reviewer 2 Report

The author have worked out on the comments and it now be accepted.

Author Response

Thanks you so much for your affirmation.